# Multi-Model Selection and Analysis for COVID-19

Nuri Ma [ID], Weiyuan Ma *[ID] and Zhiming Li

School of Mathematics and Computer Science, Northwest Minzu University, Lanzhou 730000, China;
Y200130472@stu.xbmu.edu.cn (N.M.); Y191730451@stu.xbmu.edu.cn (Z.L.)
* Correspondence: mwy2004@126.com

**Abstract:** In the face of an increasing number of COVID-19 infections, one of the most crucial and challenging problems is to pick out the most reasonable and reliable models. Based on the COVID-19 data of four typical cities/provinces in China, integer-order and fractional SIR, SEIR, SEIR-Q, SEIR-QD, and SEIR-AHQ models are systematically analyzed by the AICc, BIC, RMSE, and R means. Through extensive simulation and comprehensive comparison, we show that the fractional models perform much better than the corresponding integer-order models in representing the epidemiological information contained in the real data. It is further revealed that the inflection point plays a vital role in the prediction. Moreover, the basic reproduction numbers $R_0$ of all models are highly dependent on the contact rate.

**Keywords:** fractional order; COVID-19; multi-model selection; model evaluation

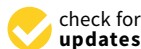

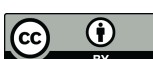

## 1. Introduction

Since December 2019, a virus named COVID-19 was first confirmed in Wuhan, China and has quickly spread throughout the world. As of 25 June 2021, the WHO reported about 179 million confirmed cases, including more than 3.8 million deaths [1]. In order to control the outbreak, it is crucial to predict the spreading trend, analyze the key epidemiological parameters, and simulate the results of various control measures. All actions of individuals and governments depend heavily on predictions of how many people will be infected. Therefore, many models have been reported to accurately estimate and predict the epidemic trend, such as empirical functions, statistical methods, dynamical and stochastic models [2–4]. Among them, integer-order and fractional dynamical models can better describe the dynamic behavior of disease transmission, which has attracted attention widely [5–7].

According to the importance of mathematical modelling, integer-order infectious disease models have been widely studied for its simplicity and accessibility. Based on the compartment hypothesis of the population, most models include compartments of susceptible (S), exposed (E), infected (I), recovered (R), death (D), quarantined (Q), asymptomatic (A), insusceptible (P), and hospitalized (H) individuals, such as SIR [8], SEIR [9], SEIR-Q [10], SEIR-QD [11], and SEIR-AHQ [12] models. In addition, many sophisticated models also have been proposed [13–16].

In the integer-order models, it is usually assumed that the infection rate, conversion rate, and removal rate are constants, which lead to the system states that are independent of history and have no memory. However, it is well known that the evolution, spread, and control of infectious diseases are closely related to the historical states, so the diseases are non-Markov processes. The more distant experience and knowledge have less influence on the present states. Thus, the growth rate of the state is not exponential, but it is more like an inverse power-law function, which can be accurately described by a fractional model. The fractional model is a generalization of the classical integer-order model. On the other hand, fractional calculus proved to be more effective in modeling dynamics with long memory and hereditary properties. The memory property of disease is embodied in integrating

more information from the past. In addition, the hereditary property reflects in the genetic profile along with evolution of the disease. All these lead to the fractional model having more advantages in modeling some biological systems than the classical integer-order. Therefore, many fractional-order models are applied to characterize memory-dependent phenomena for COVID-19, such as fractional SEIRD [17], fractional SEIQR [18], fractional SEIQRDP [19,20], fractional SEIHDR [21], and so on.

The main purposes of modeling for COVID-19 are analysis transmission trends and biological characteristics. Many dynamical models, including integer-order and fractional types, have been proposed to predict the epidemic trend. However, different models have given vastly different results of Wuhan [22–25]: the values of the basic reproduction number from 1.5 to 6, the peak times from mid-February to late March, and the total number of infections ranges from 20,000 to millions. Knowing which model is more reliable is a pending issue.

In the face of so many integer-order and fractional dynamical models, which one would be our best choice to estimate the epidemic trend and epidemiological features for COVID-19? In this paper, this pivotal problem is systematically explored by comparing the forecast ability of five models based on the four classic model selection methods. We consider integer-order and fractional SIR, SEIR, SEIR-Q, SEIR-QD, and SEIR-AHQ models for selection and analysis. Through extensive simulation and comprehensive comparison, we obtain many meaningful results, which really deepen our understanding of disease.

The remainder of the paper is arranged as follows: Section 2 introduces the integer-order and fractional models. In Section 3, four means are given for multi-model selection and analysis. Section 4 carries out comparisons and evaluations. Finally, some concluding remarks are given in Section 5.

## 2. Integer-Order and Fractional Models

We collect and study some typical integer-order models such as SIR, SEIR, SEIR-Q, SEIR-QD, and SEIR-AHQ models to find out which is the best model for COVID-19. Fractional models are constructed from the corresponding integer-order models. In these models, S represents the susceptible population, I represents the infectious population, E represents the individuals who are infected but in the latent period, P represents the isolated susceptible population, Q represents the individuals who are confirmed and infected, R denotes recovered cases, D denotes death cases, A denotes asymptomatic cases, and H denotes hospitalized cases. In the SIR and SEIR models, the parameters $\alpha$, $\gamma^{-1}$, and $\beta$ denote the infection rate, the average latent time and the cure rate, separately. In the SEIR-Q model, the parameters $\alpha_1$, $\alpha_2$, $\varepsilon$, $\gamma$, $\kappa$, $\delta$, $\lambda$, and $\eta$ denote the asymptomatic transmission rate, the symptomatic transmission rate, the onset rate, the recovery rate of infected, the testing rate, the sensitivity of test, the specificity of test, and the recovery rate of the quarantined population, respectively. In the SEIR-QD model, the parameters $\delta$, $\alpha$, $\gamma^{-1}$, $\beta^{-1}$, $\lambda$, and $\kappa$ denote the protection rate, the infection rate, the average latent time, the average quarantine time, the cure rate, and the mortality rate, separately. In the SEIR-AHQ model, the parameters $\alpha$, $\varepsilon$, $\theta$, $\sigma$, $\mu$, $\delta_1$, $\delta_2$, $\lambda$, $e$, $\omega$, $\beta_1$, $\beta_2$, and $\beta_3$ denote the transmission rate of contact, the quarantined rate, the transmissibility ratio between asymptomatic and symptomatic, the transition rate of exposed, the probability of having symptoms among infected, the transition rate of symptomatic infected, the transition rate of quarantined exposed, the rate at which the quarantined uninfected contacts are released into the wider community, the contact rate, the disease-induced death rate, the recovery rate of asymptomatic, the recovery rate of the symptomatic and the recovery rate of quarantined, respectively.

Fractional calculus is a strong tool to characterize the memory effects of the disease. The Caputo fractional derivative contains a power-law function in the convolution kernel, which can reflect the fact that the contribution of the earlier state is significantly smaller than the current state. Based on the integer-order models and the Caputo fractional derivative method [26,27], the fractional models are given as follows:

(1) Fractional SIR model

$$\begin{aligned}
{}_{C}D_{0,t}^{q}S(t) &= -\alpha S(t)I(t)/N, \\
{}_{C}D_{0,t}^{q}I(t) &= \alpha S(t)I(t)/N - \beta I(t), \\
{}_{C}D_{0,t}^{q}R(t) &= \beta I(t).
\end{aligned}$$

(2) Fractional SEIR model

$$\begin{aligned}
{}_{C}D_{0,t}^{q}S(t) &= -\alpha S(t)I(t)/N, \\
{}_{C}D_{0,t}^{q}E(t) &= \alpha S(t)I(t)/N - \gamma E(t), \\
{}_{C}D_{0,t}^{q}I(t) &= \gamma E(t) - \beta I(t), \\
{}_{C}D_{0,t}^{q}R(t) &= \beta I(t).
\end{aligned}$$

(3) Fractional SEIR-Q model

$$\begin{aligned}
{}_{C}D_{0,t}^{q}S(t) &= -(\alpha_1 E(t) + \alpha_2 I(t))S(t)/N - \kappa(1-\lambda)S(t), \\
{}_{C}D_{0,t}^{q}E(t) &= (\alpha_1 E(t) + \alpha_2 I(t))S(t)/N - (\varepsilon + \kappa\delta)E(t), \\
{}_{C}D_{0,t}^{q}Q(t) &= \kappa(1-\lambda)S(t) + \kappa\delta(E(t) + I(t)) - \eta Q(t), \\
{}_{C}D_{0,t}^{q}I(t) &= \varepsilon E(t) - (\gamma + \kappa\delta)I(t), \\
{}_{C}D_{0,t}^{q}R(t) &= \gamma I(t) + \eta Q(t).
\end{aligned}$$

(4) Fractional SEIR-QD model

$$\begin{aligned}
{}_{C}D_{0,t}^{q}S(t) &= -\alpha S(t)I(t)/N - \delta S(t), \\
{}_{C}D_{0,t}^{q}E(t) &= \alpha S(t)I(t)/N - \gamma E(t), \\
{}_{C}D_{0,t}^{q}I(t) &= \gamma E(t) - \beta I(t), \\
{}_{C}D_{0,t}^{q}Q(t) &= \beta I(t) - \lambda Q(t) - \kappa Q(t), \\
{}_{C}D_{0,t}^{q}R(t) &= \lambda Q(t), \\
{}_{C}D_{0,t}^{q}D(t) &= \kappa Q(t), \\
{}_{C}D_{0,t}^{q}P(t) &= \delta S(t).
\end{aligned}$$

(5) Fractional SEIR-AHQ model

$$\begin{aligned}
{}_{C}D_{0,t}^{q}S(t) &= -(\alpha e + e\varepsilon(1-\alpha))S(t)(I(t) + \theta A(t))/N + \lambda P(t), \\
{}_{C}D_{0,t}^{q}E(t) &= \alpha e(1-\varepsilon)S(t)(I(t) + \theta A(t))/N - \sigma E(t), \\
{}_{C}D_{0,t}^{q}P(t) &= e\varepsilon(1-\alpha)(I(t) + \theta A(t)) - \lambda P(t), \\
{}_{C}D_{0,t}^{q}H(t) &= \delta_1 I(t) + \delta_2 Q(t) - (\omega + \beta_3)H(t), \\
{}_{C}D_{0,t}^{q}Q(t) &= e\varepsilon\alpha(I(t) + \theta A(t)) - \delta_2 Q(t), \\
{}_{C}D_{0,t}^{q}I(t) &= \sigma\mu E(t) - (\omega + \gamma_1 + \delta_1)I(t), \\
{}_{C}D_{0,t}^{q}R(t) &= \beta_1 I(t) + \beta_2 A(t) + \beta_3 H(t), \\
{}_{C}D_{0,t}^{q}A(t) &= \sigma(1-\mu)E(t) - \beta_1 I(t).
\end{aligned}$$

The ${}_{C}D_{0,t}^{q}$ is the Caputo fractional derivative, which is defined as:

$$
{}_{C}D_{0,t}^{q}x(t) = \frac{1}{\Gamma(n-q)}\int_0^t (t-\tau)^{n-q-1}x^{(n)}(\tau)d\tau,
$$

where $n - 1 < q \leq n \in \mathbb{Z}^+$.

The transfer diagrams for five models are given in Figure 1. When $q = 1$, the Caputo fractional derivative reduces to the classic integer-order derivative, so the fractional SIR, SEIR, SEIR-Q, SEIR-QD, and SEIR-AHQ models become corresponding integer-order models.

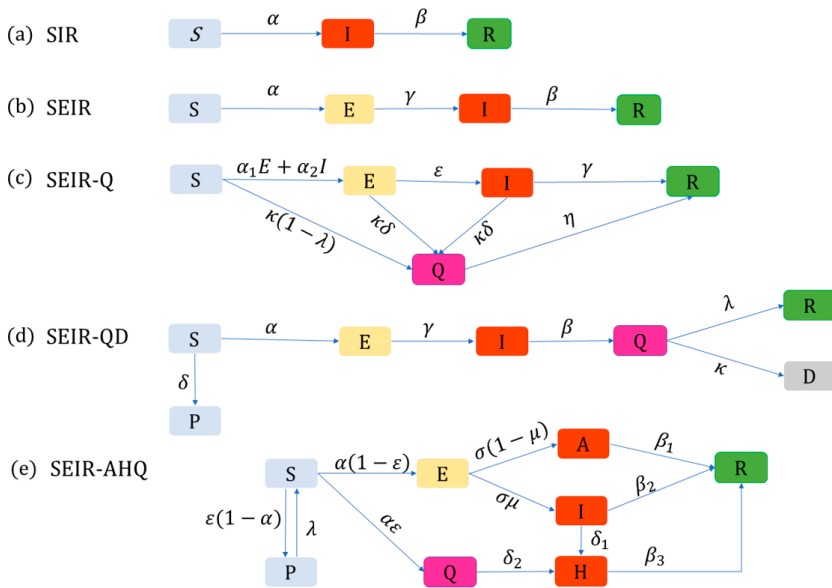

**Figure 1.** The transfer diagrams of five models. (**a**) SIR; (**b**) SEIR; (**c**) SELR-Q; (**d**) SEIR-QD; and (**e**) SEIR-AHQ.

## 3. Four Means for Model Evaluation and Analysis

In order to be closer to the disease transmission, more parameters are added, which makes the model structure more complicated. This can easily lead to the models overfitting the real data. It is not trivial to rationally evaluate various dynamical models. Furthermore, many different aspects should be considered at the same time [28], such as complexity and accuracy, fitting and prediction, robustness and sensitivity, and so on. Overemphasizing one aspect and neglecting another, consciously or unconsciously, would lead to unsatisfactory forecasts. We further introduce four means for model selection to solve this problem reasonably.

### 3.1. The Corrected Akaike Information Criterion (AICc)

The AIC was first proposed by Japanese statistician Akaike and is now a measuring standard of the degree of fitting for models. Based on the concept of entropy, the AIC can balance the model complexity and accuracy of the data fitted by the model. When $K > 40/N$, that is too many estimated parameters (K) compare to the size of the data points (N), the following corrected AIC [29] can be used:

$$AICc = Nlog(L) + 2K\frac{N}{N-K-1},$$

where N is the sample size, K is the number of parameters, and L is the likelihood function. The AICc is derived under Gaussian assumptions which are weakly dependent, so the use of AICc is highly recommended in practice.

### 3.2. The Bayesian Information Criterion (BIC)

The BIC was first introduced by Schwarz (1978) as a competitor to AIC. The BIC is an asymptotic approximation to a transformation of the Bayesian posterior probability of the model [30], which is defined as:

$$BIC = Nln(L) + kln(N).$$

The smaller value of BIC means better fitting accuracy between model and data. Compared with AIC, the BIC punishes the model parameter number more in the case of large data; therefore, the BIC prefers to choose simple models with fewer parameters.

### 3.3. The Root Mean Square Error (RMSE)

The RMSE is used to calculate how much the forecasted value differs from the actual value [31]. The smaller value shows the higher prediction accuracy; therefore, the root mean square error is used as an evaluation standard of measurement precision. The RMSE is defined as:

$$RMSE = \sqrt{\frac{\sum_{i=1}^{N}(A_i - F_i)^2}{N}},$$

where $N$ is the size of data points, and $A_i$ and $F_i$ represent the actual value and forecasted value, separately. The RMSE can be applied to estimate the accuracy of models.

### 3.4. The Pearson's Correlation Coefficient (R)

In order to accurately describe the degree of correlation between the forecasted value and the actual value [31], the Pearson's correlation coefficient ($R$) is defined as:

$$R = \frac{N\sum_{i=1}^{N} A_i F_i - (\sum_{i=1}^{N} A_i)(\sum_{i=1}^{N} F_i)}{\sqrt{N(\sum_{i=1}^{N} A_i^2) - (\sum_{i=1}^{N} A_i)^2}\sqrt{N(\sum_{i=1}^{N} F_i^2) - (\sum_{i=1}^{N} F_i)^2}}.$$

The value of $R$ satisfies $-1 < R < 1$. The $+$ and $-$ show positive and negative linear correlations between the forecasted and actual values, respectively. The $R$ greater than or equal to 0.8 is considered as strong correlation. $R^2$ demonstrates the strength of the association between $A_i$ and $F_i$.

## 4. Model Evaluation and Analysis Based on the Real Data

To make a quantitative comparison of different models, the outbreak of COVID-19 is studied, which has seriously affected economy and health of people around the world. In this section, we apply four means to evaluate and analyze the most widely used models in the field of epidemiology. Since the data link of COVID-19 in China is intact, we use the data from the China National Health Commission (http://www.chinacdc.cn/, accessed on 12 September 2021) for simulation. The fifty-four-day data from 22 January 2020 to 15 March 2020 of Beijing, Chongqin, Tianjing, and Heilongjiang are used, which are considered as some successful cities during the battle against the disease in China. The unknown parameters involved in integer-order and fractional models are identified by the non-linear least squares method and Simulink Design Optimization of MATLAB. Through a large number of numerical tests, the following conclusions can be drawn.

### 4.1. The Fractional Models Can Better Fit the Real Data than the Corresponding Integer-Order Models

It is well known that the outbreak of COVID-19 is not only dependent on the number of current infected people, but also relates to those who were infected in the past; that is, the disease has memory. The fractional epidemic model can reflect the memory and heredity properties, which is more compatible with the spread process of disease. The four means are computed to evaluate and analyze the validity of integer-order and fractional models.

The smaller AICc, BIC, RMSE values and the larger R value of model indicate the better fitting accuracy, respectively. The difference of four means between integer-order and fractional models are given as:

$$D_j = INT_j - FRA_j,$$

where $j$ is the AICc, BIC, RMSE, or R, and $INT_j$ and $FRA_j$ are the values of the mean j on the integer-order model and the fractional model, respectively. When $D_{\text{AICc}} > 0$, $D_{\text{BIC}} > 0$, $D_{\text{RMSE}} > 0$, and $D_R < 0$, the fractional model has better fitting accuracy than the corresponding integer-order model, respectively.

In Figure 2, the $D_j$ are calculated from 22 January 2020 to different end times in Beijing, Chongqing, Tianjin, and Heilongjiang. It is found that all of the $D_j$ about AICc, BIC, and RMSE are greater than 0, and the $D_R$ values are less than 0; that is, the fractional models are better than the corresponding integer-order models for different time windows.

For convenience, all of our subsequent discussions are based on the fractional models.

### 4.2. No Model Is Reliable for Long-Term Forecasting Based on the Early-Stage Real Data

The rapid spreading of COVID-19 results in a dramatic increase in the number of infections. Therefore, how to predict its spread trend is a tough task. Five fractional models are used to estimate long-term prediction ability. In Figure 3, the predication and fitting results of three cities (Beijing, Chongqing, Tianjin) and Heilongjiang province are given by AICc, BIC, RMSE, and *R* means. In the early stage of the epidemic with very limited data, all models can fit and predict spread trend in a short-term very well, but no model is reliable for long-term forecasting. All models always underestimate or overestimate the size of the epidemic for long-term prediction.

### 4.3. The Fractional SEIR-Q and SEIR-QD Models Can More Accurately Describe the COVID-19 Spread Trends

Fractional dynamical models are more in line with the biological significance by specific parameters (fractional order). The fractional dynamical models can be divided into three groups based on their performance. In Figure 4, the fractional SIR and SEIR models look insufficient to describe the spread trends, especially in the final stage, since too few parameters are involved. The fractional SEIR-AHQ model contains redundant parameters, which generates the larger value of AICc. That leads to its fluctuation being the greatest among the models. The fractional SEIR-Q and SEIR-QD systems are more suitable to characterize the infectious characteristics of COVID-19 by appropriately introducing the effect of quarantine.

### 4.4. The Inflection Point of the Real Data Is Vital for Prediction

The inflection point plays a pivotal role in predicting the epidemic trend, which is in agreement with previous report [28]. In Figure 3, when enough data are available, typically when the number of infected people passes the inflection point, forecast values converge to the true values by fractional models. For example, all models fail to make a convincing forecast in Beijing when using the data before 14 February (early stage and middle stage), which is also the inflection point. However, after the inflection point (late stage), fractional SEIR-Q, SEIR-QD, and SEIR-AHQ models can successfully capture the real data.

### 4.5. A Single Mean Is Insufficient to Evaluate the Model's Prediction Capability

When using the models to fit data from the epidemic process, four means are used to select suitable models. The AICc and the BIC consider the trade-off between the fitting effect of model and the principle of least parameters, but the AICc is more sensitive to the parameters. The RMSE measures the degree of deviation between the real data and the fitted data, and the *R* highlights the correlation between the outcomes of model and the real data. It should be note that a single evaluation criterion is not reliable. For example, in Figure 4a, when the end times are from 4 February to 8 February, the most suitable model is the fractional SIR, but the fractional SEIR-QD model achieves the better trade-off in the rest time period. In Figure 4b, the fractional SEIR-QD model can be picked out by BIC. In Figure 4c, the fractional SEIR-AHQ model is the best model by RMSE. In Figure 4d, the fractional SEIR-Q, SEIR-QD, and SEIR-AHQ models are considered as strong correlations between the forecasted data and the actual data by *R*. Therefore, the models cannot be evaluated by a single mean; namely, the means should be combined to select suitable models.

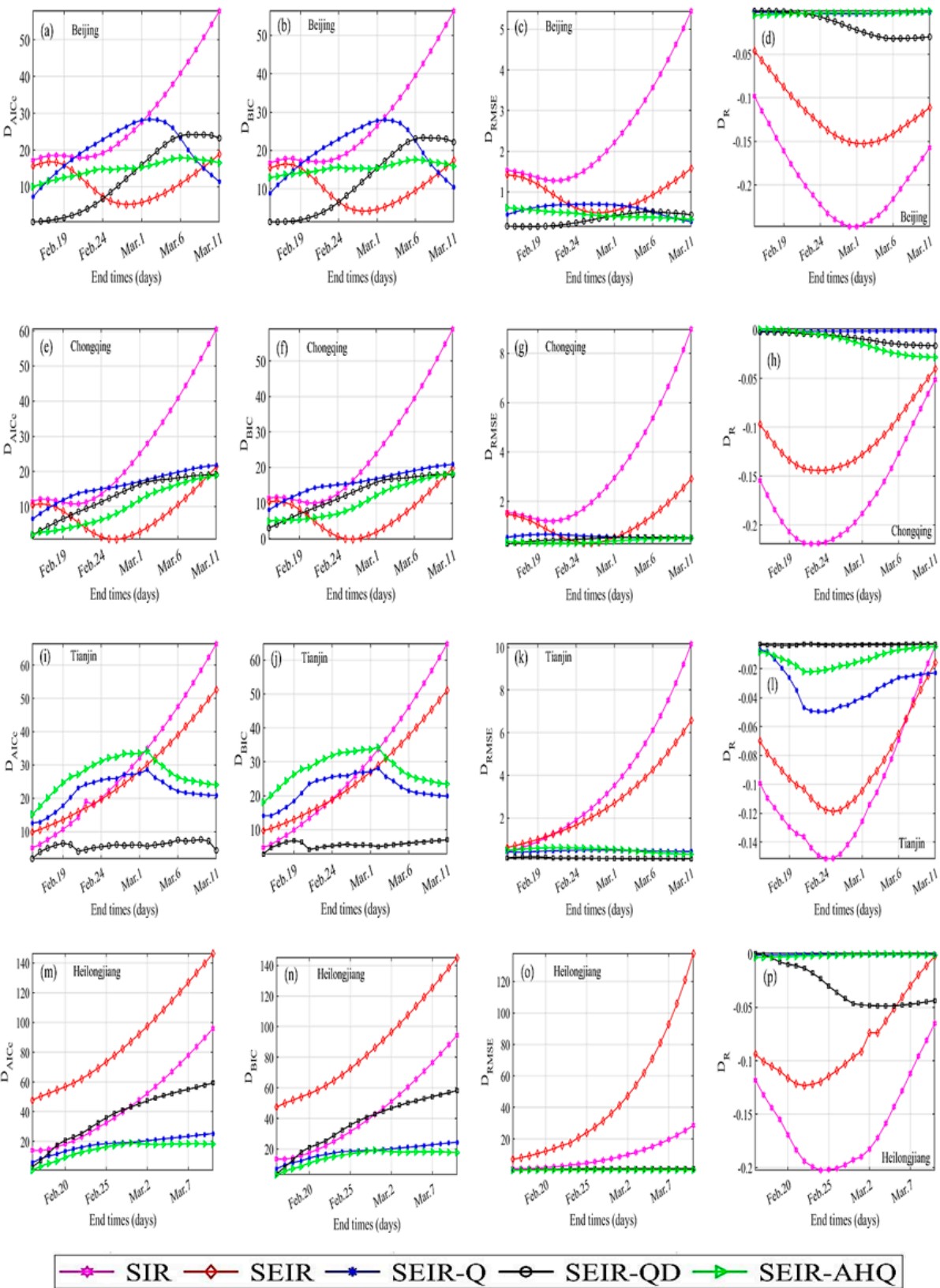

**Figure 2.** The difference values $D_j$ from 22 January 2020 to different end times in Beijing, Chongqing, Tianjin, and Heilongjiang. (**a**–**d**) $D_j$ of Beijing; (**e**–**h**) $D_j$ of Chongqing; (**i**–**l**) $D_j$ of Tianjin; (**m**–**p**) $D_j$ of Heilongjiang.

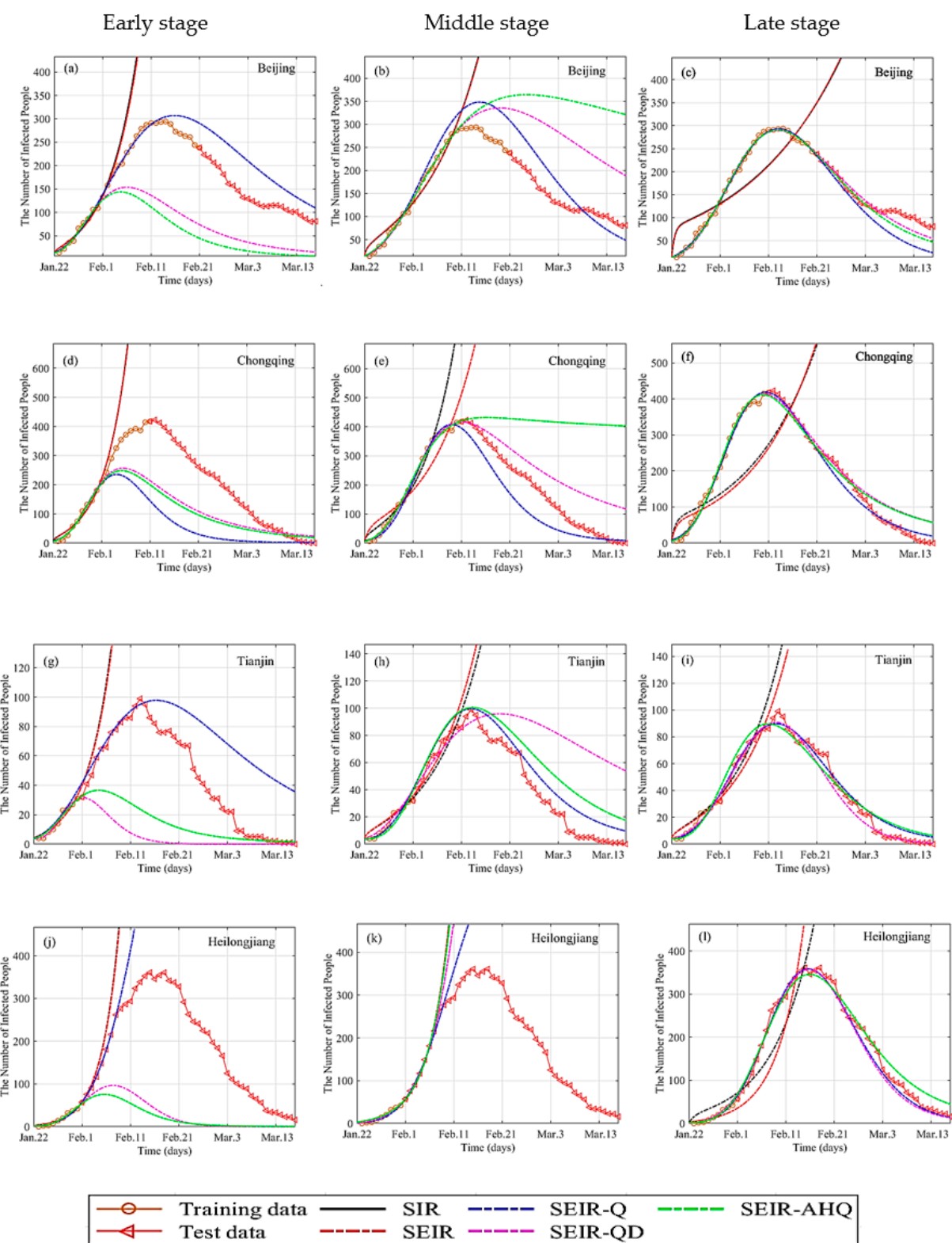

**Figure 3.** Prediction of the COVID-19 epidemic trends in Beijing city, Chongqing city, Tianjin city, and Heilongjiang province from 22 January 2020 to 5 March 2020 based on the data of first 10 (early stage), 20 (middle stage), and 30 (late stage) days, respectively. (**a**–**c**) The number of infected people in Beijing at different stages; (**d**–**f**) The number of infected people in Chongqing at different stages; (**g**–**i**) The number of infected people in Tianjin at different stages; (**j**–**l**) The number of infected people in Heilongjiang at different stages.

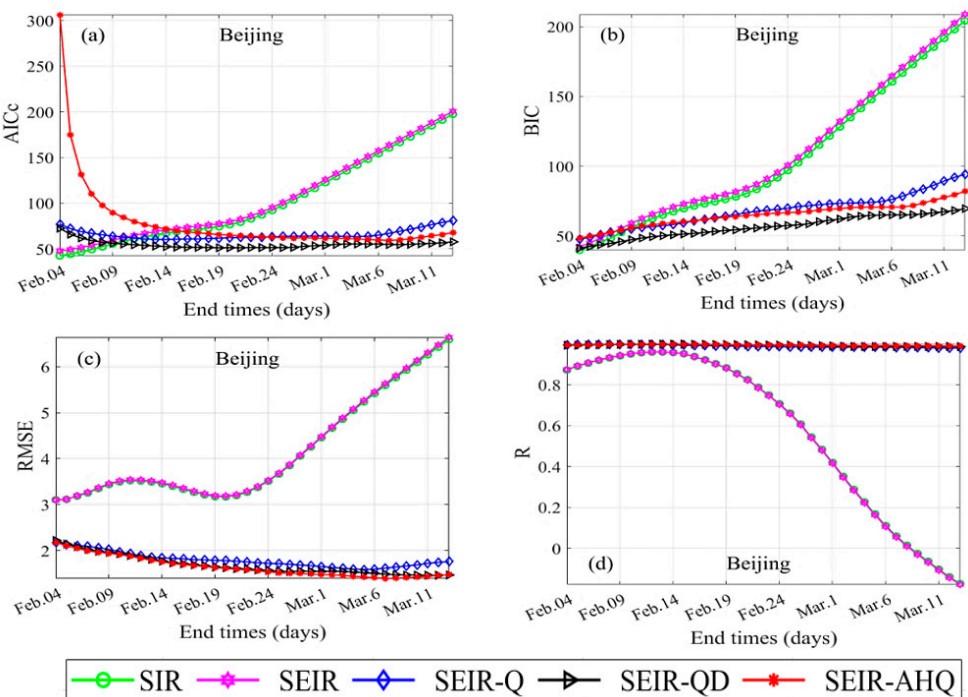

**Figure 4.** The values of AICc, BIC, RMSE, and R from 22 January to different end times in Beijing. (**a**) AICc; (**b**) BIC; (**c**) RMSE; (**d**) R.

### 4.6. All of the Basic Reproduction Number $R_0$ Heavily Depend on Contact Rate

The sensitivity analysis for the fractional epidemic models is to study the main intervention measures that affect the COVID-19 dynamics. Using the next-generation matrix method [32], the basic reproduction number of fractional SIR, SEIR, SEIR-Q, SEIR-QD, and SEIR-AHQ models are given as $\frac{\alpha}{\beta}$, $\frac{\alpha}{\beta}$, $\frac{\alpha_1(\gamma+\kappa\delta)+\alpha_2\varepsilon}{(\varepsilon+\kappa\delta)(\gamma+\kappa\delta)}$, $\frac{\alpha}{\beta}$ and $\frac{\beta_1\alpha\mu(1-\varepsilon)+\alpha(\beta_2+\delta_1)(1-\varepsilon)(1-\mu)}{\beta_1(\beta_2+\delta_1)}$, respectively. Latin hypercube sampling (LHS) and partial rank correlation coefficient (PRCC) [33] are used to analyze the influence of the parameters on the model. The higher the PRCC value of a parameter, the greater its impact on the dynamic behavior [34]. The involved parameters are given in Table 1, which includes PRCC and *p*-values. The bar graph of PRCC results is shown in Figure 5. Furthermore, whatever the model, $R_0$ are heavily dependent on the contact rate $\alpha$, except for $\alpha_1$ and $\alpha_2$ in the SEIR-Q model. It is obvious that non-medical intervention measures (social distance restriction, self-isolation, staying at home or wearing masks, etc.) have directly impacted on epidemic prevention and control.

Based on the references [6,20,21,35], we could easily obtain unique solutions and a locally asymptotically stable of equilibrium points. We only take fractional SEIR-QD as an example; other models can get similar results.

**Remark 4.1.** *The initial value problem for the fractional SEIR-QD model has a unique solution in $\mathbb{R}_+^7$.*

**Remark 4.2.** *The diseases-free equilibrium (0, 0, 0, 0, $R^*$, $D^*$, $P^*$) of the fractional SEIR-QD system is locally asymptotically stable if $R_0 < 1$ and unstable if $R_0 > 1$.*

**Table 1.** The PRRC values of $R_0$ with corresponding *P*-values.

| Models | Parameters | PRCC Values | *p*-Values |
|---|---|---|---|
| SIR | $\alpha$ | 0.8866 | 0.0000 |
| | $\beta$ | −0.4624 | $1.5497 \times 10^{-106}$ |
| SEIR | $\alpha$ | 0.8667 | 0.0000 |
| | $\beta$ | −0.4556 | $4.3039 \times 10^{-103}$ |
| | $\alpha_1$ | 0.7326 | 0.0000 |
| | $\alpha_2$ | 0.2386 | $2.7566 \times 10^{-27}$ |
| SEIR-Q | $\varepsilon$ | −0.5570 | $2.1135 \times 10^{-163}$ |
| | $\kappa$ | −0.2734 | $1.3051 \times 10^{-35}$ |
| | $\delta$ | −0.1798 | $5.4878 \times 10^{-16}$ |
| | $\gamma$ | 0.0396 | 0.0763 |
| SEIR-QD | $\alpha$ | 0.8823 | 0.0000 |
| | $\beta$ | −0.4399 | $1.9184 \times 10^{-95}$ |
| | $\alpha$ | 0.7213 | $5.2370 \times 10^{-321}$ |
| | $\mu$ | −0.5645 | $1.0293 \times 10^{-168}$ |
| SEIR-AHQ | $\varepsilon$ | −0.3493 | $1.7440 \times 10^{-58}$ |
| | $\delta_1$ | −0.1348 | $1.4459 \times 10^{-9}$ |
| | $\beta_1$ | −0.0727 | 0.0011 |
| | $\beta_2$ | 0.0207 | 0.3542 |

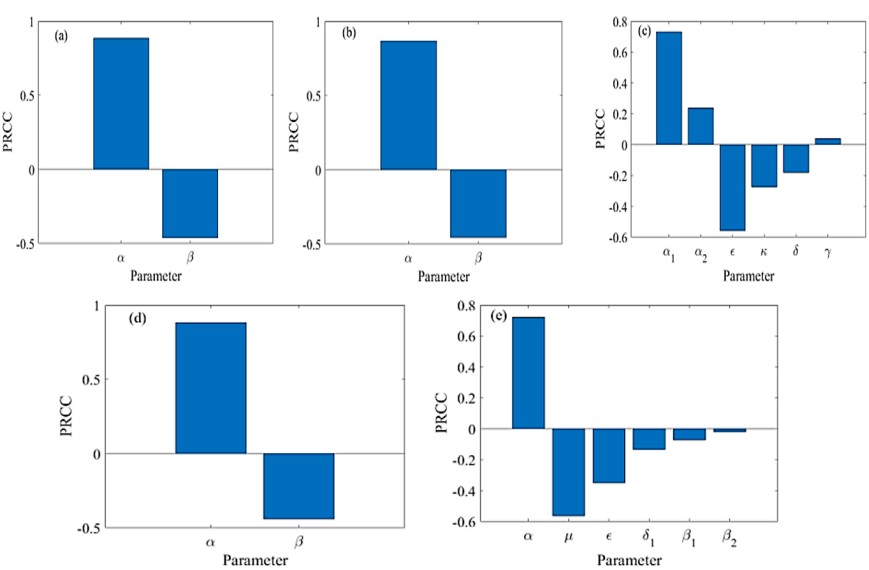

**Figure 5.** The PRCC results of parameters involved in $R_0$. (**a**) SIR; (**b**) SEIR; (**c**) SELR-Q; (**d**) SEIR-QD; (**e**) SEIR-AHQ.

## 5. Conclusions

In this paper, we systematically studied the prediction capability of five widely used integer-order and fractional-order models with the COVID-19 data in China. Through numerical simulations, many insightful results are obtained by the four means and the PRCC. Firstly, the fractional models have better performance than the corresponding integer-order models. Fractional models can better describe epidemic characteristics in the diseases by an additional parameter (fractional order). Secondly, almost all of the models overestimate or underestimate the infected people of outbreak based on the early-stage data. Thirdly, the fractional SEIR-Q and SEIR-QD models can better predict the scales of epidemic trends, which have the better trade-off between the model complexity and the fitting accuracy. Fourthly, it is found that the inflection points in the data play a very important role. The prediction results of all integer-order and fractional models are closely relate to inflection points. Fifthly, a single evaluation criterion is insufficient to estimate the model's capability. The best model for modeling COVID-19 can be picked out by AICc

and BIC means. Finally, based on the LHS and PRCC methods, we find that non-medical intervention measures play a critical role in COVID-19 control.

This paper provides a new insight about selection and analysis of infectious diseases by fractional modeling. Furthermore, models with vaccine can play an important role in the allocation of vaccines, which are not considered in this paper. We will discuss this in our future work.

**Author Contributions:** N.M.: Prepared the draft. W.M.: Developed the concept, methodology, and revised the paper finally. Z.L.: Performed the analysis. All authors have read and agreed to the published version of the manuscript.

**Funding:** This work was supported by the Fundamental Research Funds for the Central Universities (Grant No. 31920210018), and the Innovation Team of Intelligent Computing and Dynamical System Analysis and Application of Northwest Minzu University.

**Institutional Review Board Statement:** Not applicable.

**Informed Consent Statement:** Not applicable.

**Data Availability Statement:** Not applicable.

**Conflicts of Interest:** The authors declare no conflict of interest.

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
