# Peer review of "Multi-Model Selection and Analysis for COVID-19"

_fractalfract, doi:10.3390/fractalfract5030120_

Round 1

Reviewer 1 Report

The aim of this paper is to focus on the integer-order and fractional-order SIR, SEIR, SEIR-Q, SEIR-QD and SEIR-AHQ models of COVID-19. The cases study is China. By a comparative study the authors have tried to show the ability and efficiency of the study to show that the fractional models perform much better than the corresponding integer-order models in representing the epidemiological information contained in the data. I have studied the paper and this paper can be published after the following revisions:

1- I believe that the introduction can be improved by adding more background. Please cite the following references: New Procedures
of a Fractional Order Model of Novel coronavirus (COVID-19)
Outbreak via Wavelets Method. Axioms 2021, 10, 122. https://doi.org/
10.3390/axioms10020122 and A Novel Technique to Control the Accuracy of a Nonlinear Fractional Order Model of COVID-19: Application of the CESTAC Method and the CADNA Library. Mathematics 2021, 9, 1321. https://doi.org/
10.3390/math9121321

2- Please add some new sentences to clarify the physical meaning of fractional order derivative. 

3- In the second section's title, please change "Fractiona" to "Fractional". 

4- After models you have to describe all parameters and functions. 

5- The stability of the model and existence of solution must be added. At least you should cite some references. 

6- It not clear that the authors have applied the simulated data or real!!

7- What is the order q in the graphs? 

I propose a major revision.  

Reviewer 2 Report

All comments and suggestions are presented in the attached file. 

Round 2

Reviewer 1 Report

I do not have more comments on this paper. 

Reviewer 2 Report

I checked the revised version very carefully; I recommend it for publication in its present form